# The content and completeness of women-held maternity documents before admission for labour: A mixed methods study in Banjul, The Gambia

**Lotta Gustafsson**[1☯‡], **Fides Lu**[1☯‡], **Faith Rickard**[1], **Christine MacArthur**[2], **Carole Cummins**[2], **Ivan Coker**[3], **Kebba Mane**[3], **Kebba Manneh**[4], **Amie Wilson**[2], **Semira Manaseki-Holland**[2]*

**1** University of Birmingham Medical School, Edgbaston, Birmingham, United Kingdom, **2** Institute of Applied Health Research: University of Birmingham, Birmingham Clinical Trials Unit College of Medical and Dental Sciences, Birmingham, United Kingdom, **3** Bundung Maternal and Child Health Hospital, Banjul, The Gambia, **4** Kanifing General Hospital, Banjul, The Gambia

☯ These authors contributed equally to this work.
‡ First Authors
* S.ManasekiHolland@bham.ac.uk

**Data Availability Statement:** Data cannot be shared publicly because the data could indirectly identify participants. Although we recognise data-

## Abstract

### Background

Women-held maternity documents are well established for enabling continuity of maternity care worldwide, with the World Health Organisation (WHO) recommending their use in effective decision-making. We aimed to assess the presence, content and completeness of women-held maternity documents at admission to hospitals in The Gambia, and investigate barriers and facilitators to their completion.

### Methods

We interviewed 250 women on maternity wards of all 3 Banjul hospitals and conducted content analysis of documentation brought by women on admission for their completeness against WHO referrals criteria. Logistic regression models were used to estimate the odds of the minimum criteria being met. Two focus groups and 21 semi-structured interviews (8 doctors, 8 midwives and 5 nurses) were conducted with healthcare practitioners to explore barriers and facilitators to documented clinical information availability on admission.

### Findings

Of the women admitted, all but 10/250 (4%) brought either a maternity card or a structured referral sheet. Of all forms of documentation, women most frequently brought the government-issued maternity card (235/250, 94%); 16% of cards had all 9 minimum criteria completed. Of the 79 referred women, 60% carried standardised referral forms. Only 30% of 97 high-risk women had risk-status recorded. Women were less likely to have documents complete if they were illiterate, had not attended three maternity appointments, or lived more

sharing is an important principle underpinning scientific research, patient confidentiality was a clear part of our ethical approval processes. Our quantitative data includes sensitive information such as participant's socioeconomic background, occupation, age, number of children, and ethnicity. Similarly the qualitative transcripts in full would allow identification of participants. However, relevant parts of the transcripts can be made available on request and we would be very willing to consider requests for the quantitative data. The data are available from the Internal Research Ethics Committee at the University of Birmingham (contact via email: posh-irec@contacts.bham.ac.uk) for researchers who met the criteria for access to confidential data. The name of the data sets to request are: 1. Women held documents in The Gambia - admission (quantitative) 2. Women held document in The Gambia (qualitative transcripts)

**Funding:** This work recieved funding from the University of Birmingham Intercalation Award: Arthur Thompson Trust Award to LG. The funder had no role in study design, data collection and analysis, decision to publish, or preparation of the manuscript.

**Competing interests:** The authors have declared that no competing interests exist.

than one hour from hospital. During qualitative interviews, three themes were identified: women as agents for transporting information and documents (e.g. remembering to bring maternity cards); role of individual healthcare professionals' actions (e.g. legibility of hand-writing); system and organisational culture (e.g. standardised referral guidelines).

## Conclusion

Women rarely forgot their maternity card, but documents brought at admission were frequently incomplete. This is a missed opportunity to enhance handover and quality of care, especially for high-risk women. National guidelines were recognised by providers as needed for good document keeping and would enhance the women-held maternity documents' contribution to improving both safety and continuity of care.

## Background

Reducing maternal mortality is a high priority on the international health agenda[1]. In 2013, The Gambia had a maternal mortality ratio (MMR) of 433 per 100,000 live births, one of the highest globally[2,3]. Attempts to reduce maternal mortality in line with the Sustainable Development Goals are challenging in this resource-limited setting; delays or errors to decision-making processes due to an inadequate maternity history or case documentation contribute to these maternal deaths[4–6]. The World Health Organisation(WHO)'s 2016 'Standards for Improving Quality of Maternal and Newborn Care in Health Facilities' emphasises that increasing maternal health facility coverage is not enough and that improving the quality of information and referral systems, specifically documentation, is one of the keys to improving outcomes[7].

Women-held documents are well established in maternity care worldwide; at least 163 countries are known to use some form of home-based record[8–11]. Based on research, the WHO continue to emphasise their use to help in decision-making and 'continuity, quality of care and [a mother's] pregnancy experience'[11–16]. Additionally there is good evidence that women-held documents have other advantages, including that they help women to feel empowered and more involved in their own health and that of their babies[16–19].

Women-held documents often take the form of maternity cards. The maternity card has the potential to assist in continuity of care within maternity services as it enables the handover of clinical information, between antenatal appointments, at admission to maternity units and during post-natal care[11,14,19]. As women frequently move from one facility to another during their pregnancy in both low and high-income countries, women-held documents can ensure the clinical history is available and so pregnancy complications are more likely to be detected and acted upon[14,16,18,19,20,21]. A WHO collaborative study in 1993 specifically highlighted the potential use of maternity cards to enhance the diagnosis and referral of high-risk women[16]. However, to fulfil this function, the cards must be consistently brought to appointments by women, looked at by staff and filled out in their entirety with contents that fulfil WHO recommendations.

Whilst the maternity card is recommended, its feasibility and use in a real-world (especially resource limited) setting is important to establish. The maternity card is a low-cost resource already widely accepted across low-income countries (LICs)[11,20] and research has suggested that women rarely forget their documents at antenatal appointments[16,22,23]. However,

systematic reviews state that no studies have assessed the availability of antenatal records at the time of delivery (rather than between antenatal appointments)[22] or reported on the content or completeness of maternity women-held documents at the time of birth[18,19]. This is arguably the most important time when antenatal handover information is required.

Establishing the effectiveness and use of maternity card use in The Gambia may yield findings relevant to other LICs and countries in the region that face similar challenges to maternal health. This study further addresses research gaps laid out by both the WHO and recent systematic reviews[11,18,19].

The primary aim of this study was to assess the number, type, content quality and completeness of women-held documents on admission to maternity units in The Gambia, a LIC with a high MMR[2]. Secondly, we aimed to explore context-specific barriers and facilitators to effective use of women-held documents in maternity units by health professionals and maternity staff, especially for women admitted with high-risk pregnancies or deliveries.

## Methods

This was a convergent parallel mixed-methods study[24] that took place across all three maternity hospital departments in the Greater Banjul and Kanifing region between January and March 2018 (S1 Table). Antenatal services are well attended in The Gambia with over 90% visiting a clinic at least once during their pregnancy at a variety of health facilities ranging from mobile health posts and local health centres to the tertiary hospital in Banjul[25] The government-issued yellow maternal card is an A4 double-sided piece of card supplied on the first antenatal appointment and intended to stay with the woman until her final post-natal check-up (Figs 1 and 2).

### Quantitative data

In-patient women on antenatal, postnatal and maternity high-dependency units, aged 16 and over at the three study hospitals were invited to participate whilst waiting to leave the ward after formal 'discharge'. Researchers rotated around the three hospitals across an even distribution of days throughout the study period (including weekends) and between each facility to increase the likelihood of obtaining a representative sample during the study period and to reduce any differences between observers. Researchers were present throughout the period of the day that discharges took place (between 9am and 1pm) and recruited all eligible women discharged that day who gave informed consent (thumbprint or signature). Women taking part in any Medical Research Council (MRC) study (n = 21; to comply with local MRC ethical approval requirements) and women unable to speak English, Mandinka, Wolof or Fula (n = 0) were excluded. Ethical approval was granted by the Scientific Coordinating Committee, the joint Government/MRC Ethics Committee in The Gambia and the University of Birmingham BMedSc Population Sciences and Humanities Internal Research Ethics Committee.

To achieve a 95% confidence level ($\alpha$ = 0.05) with a ±5% accuracy, a minimum sample size of 243 was required to estimate the number of women who brought a maternity card, based on a population of an unknown size[26]. This was assuming that an 80% prevalence of women would bring their maternity card to the maternity unit, based on our unpublished research in Kerala, India[27].

A verbal questionnaire, adapted and piloted from studies in Mongolia[28] and India[29], was administered on wards with the help of trained local interpreters (S1 Text). The researcher also conducted a document review by recording type and contents of any documentation brought to or received from the ward (including the maternity card) and ward-based patient medical records were reviewed to establish the reason for women's admission.

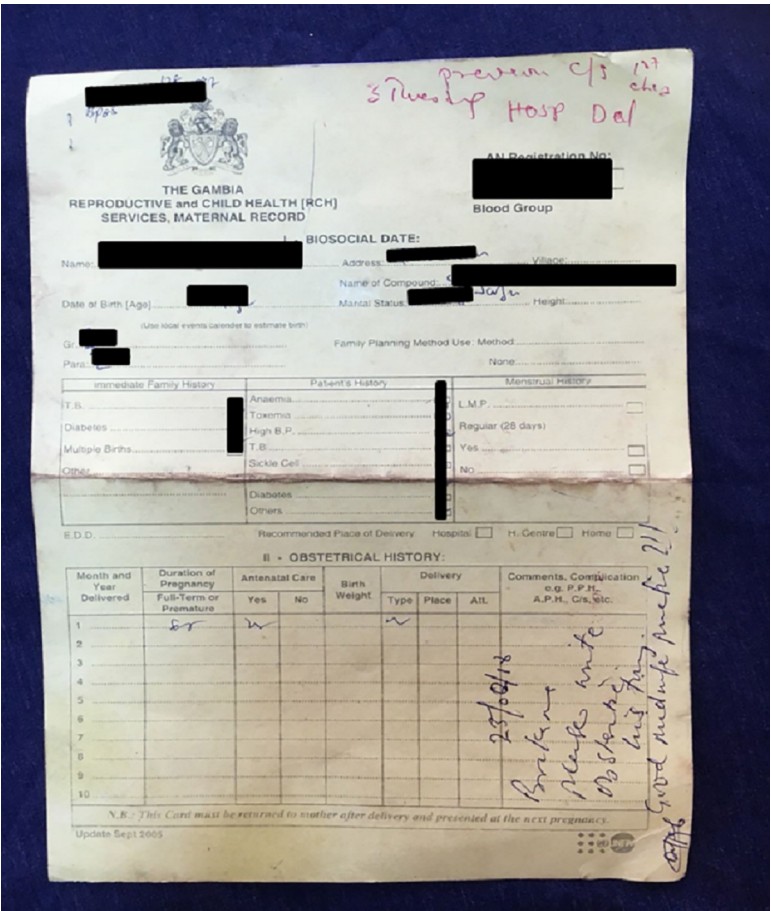

**Fig 1. The front page of the government issued women-held 'yellow Maternity Card'.** Risk status was sometimes added in red ink at the top of the card. A midwife has written a warning about document completeness on this particular card.

Content quality and completeness was determined by comparison with minimum criteria from WHO recommendations for maternity referrals that focus on emergency situations (Fig 3)[30] and analysed as median number of criteria fulfilled (criteria fulfilled: yes or no). Although not all admissions were under emergency situations, these criteria were deemed appropriate on consultation with UK/Gambian maternal health experts, including the UK Royal College of Obstetrics and Gynaecology. Additional items considered of interest and those that had a designated space on the maternity card were assessed but not included in the minimum criteria analysis.

Data was analysed using SPSS 24.0 (IBM, Armonk, NY, USA). Descriptive statistics (numbers and percentages) characterised the nature and quality of documents; including the number of each type of document, whether individual criteria were met and how many women carried documents that met the minimum criteria. Subgroups of women who were referred, attended scanning, or were high-risk were explored. Percentages were calculated using non-missing data as the denominator.

To establish whether any particular characteristics could predict whether the minimum criteria would or would not be met by a woman's documents, binomial logistic regression models were used to produce both unadjusted and adjusted odds ratios. Predictor variables were entered into the model based on clinical rationale and on findings from the qualitative arm of

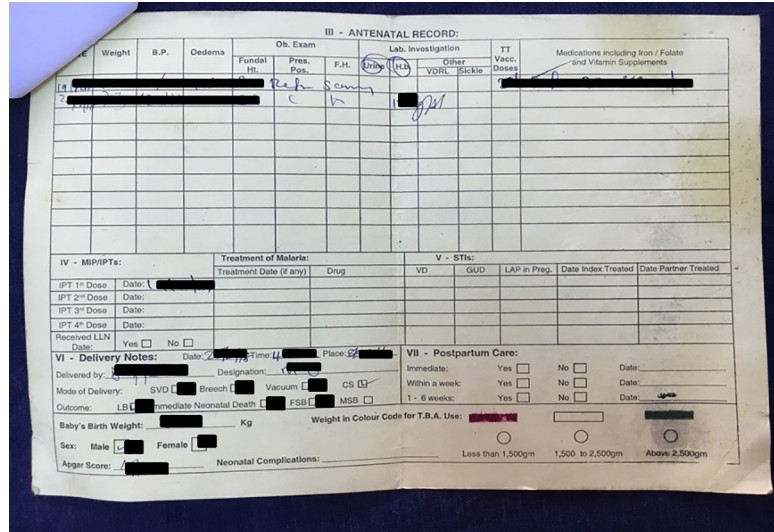

**Fig 2. The inside of the government issued women-held 'yellow Maternity Card'.** Generally this side of the card contains information relevant post-discharge.

the study. The rationale was supported by evidence from Mongolia and India that low socio-economic status and living far from the hospital can lower the quality of their written documentation[11,28,29]. Occupation was regarded as the best representation of socio-economic status (preferable to house structure as most women lived in cement and corrugate property). Therefore; age, occupation, time to get to hospital, number of children, English literacy, whether they had been referred, and number of previous contacts were all deemed to be

**Minimum Criteria Analysis for referral, adapted from WHO criteria[30]**

1. Name
2. Age
3. Address
4. Parity
5. Gestational Age (or estimated date of delivery)
6. Complications in antenatal period
7. Relevant past obstetric complications
8. Treatments applied so far
9. Results of those treatments

**Fig 3. Minimum criteria.** Minimum criteria for referral pattern improvement were adapted from WHO criteria[30] after consultation with maternal health experts at the UK Royal College of Obstetrics and Gynaecology. Criteria are applied to all admissions on the inference that the transfer of care from antenatal services to maternity unit admission is a form of 'referral'.

potentially associated with document completeness. Address, risk status and transport to hospital were initially included but were removed from the model due to multi-collinearity in association with other variables, as revealed through examining the correlation matrices and cross-tabulations. 'Hospital 1, 2 or 3' was also added to the regression model as a fixed variable to adjust for any hospital effects.

### Qualitative data

In order to establish barriers and facilitators to effective use of maternity cards and documentation, qualitative and quantitative components of the study were conducted in parallel in all 3 hospitals. A purposive sampling frame included all cadres of healthcare professionals (HCPs) including nurses, midwives and doctors (see S1 Table for numbers of staff who work at each hospital). Initially, we planned semi-structured interviews (SSIs) to obtain in-depth and sensitive views and experiences which may be difficult to elicit in a group setting and focus group discussions (FGDs) to explore more in-depth group dynamics, agreed behaviours and ways of operation. However, the FGDs proved difficult to arrange due to staff availability; it was seemingly rare for large numbers of staff to be available at the same time. For the SSIs, sampling was much more systematic. Participants were allocated to either FGDs or one-to-one SSIs; no staff participated in both. Participants were recruited until thematic saturation was achieved. FDGs and SSIs were conducted in English (all HCPs spoke fluent English) at the hospital sites using a pre-determined topic guide (S2 Text) that addressed views on current practices of information exchange, barriers and facilitators to effective handover using the documents, and feasible opportunities for improvement. The FDGs and SSIs were recorded, then transcribed verbatim and anonymised by the researcher.

Inductive thematic analysis based on Braun and Clarke's six-step approach[31] was undertaken to identify themes from the data. The researcher performed line-by-line coding on all transcripts and another researcher independently coded four of the most data-rich transcripts for analyst triangulation[32]. Themes and subthemes were subsequently developed and refined. Convergent triangulation was used to combine quantitative and qualitative results in the discussion.

Due to resource and time constraints, it was not possible to conduct qualitative research on the women's perspectives of the maternity card process for handover by HCPs. The authors collected qualitative data concerning the major barriers or facilitators to the effective completion and use of the card by HCPs rather than women's views of health-care providers use of the card.

## Results

### Quantitative component results

In total, 251 eligible women were approached. None refused, but one lacked capacity to consent hence was not included. 250 women completed the questionnaire in the study period. This represented approximately 25% of all discharges from the three facilities throughout the study period (total 1,082; Hospital 1–193, Hospital 2–371, Hospital 3–518). Table 1 shows women's demographic and admission characteristics.

On arrival at the maternity department, documents brought by women included: maternity cards (Figs 1 and 2), structured referral sheets (Fig 4), and a selection of less frequently presented documents (ultrasound reports, prescription notes, scraps of paper, child health reports, miscellaneous lab requests/results). All but 10/250 (4%) of women had brought either a structured referral sheet or a maternity card.

**Table 1. Women's demographic and admission characteristics across the three hospitals study sites.**

| Background Characteristic | Hospital 1 (n = 72) | Hospital 2 (n = 91) | Hospital 3 (n = 87) | Total (n = 250) |
|---|---|---|---|---|
| | No. (%) | No. (%) | No. (%) | No. (%) |
| **Age**[*] | | | | |
| 20 and under | 11 (15.3) | 16 (17.6) | 17 (19.5) | 44 (17.6) |
| 21–29 | 31 (43.1) | 38 (41.8) | 45 (51.7) | 114 [*]45.6) |
| 30 and over | 30 (41.7) | 37 (40.7) | 25 (28.7) | 92 (36.8) |
| **Occupation**[*] | | | | |
| Housewife | 48 (66.7) | 52 (57.1) | 62 (71.3) | 162 (64.8) |
| Retail | 12 (16.7) | 26 (28.6) | 11 (12.6) | 49 (19.6) |
| Other[a] | 23 (16.7) | 13 (14.3) | 14 (16.1) | 39 (15.6) |
| **Time taken to get to hospital**[*] | | | | |
| Under 1 hour | 43 (61.4) | 58 (63.7) | 86 (98.9) | 187 (75.4) |
| 1 hour and above | 27 (38.6) | 33 (36.3) | 1 (1.1) | 61 (24.6) |
| **Transport used to get to hospital** | | | | |
| Walked | 5 (6.9) | 1 (1.1) | 8 (9.2) | 14 (5.6) |
| Taxi / Gelli | 31 (43.1) | 76 (83.5) | 79 (90.8) | 186 (74.4) |
| Ambulance | 36 (50.0) | 14 (15.4) | 0 (0.0) | 50 (20.0) |
| **Number of children**[b] [*] | | | | |
| 0 | 3 (4.2) | 6 (6.6) | 3 (3.4) | 12 (4.8) |
| 1–2 | 36 (50.0) | 44 (48.4) | 47 (54.0) | 127 (50.8) |
| 3–4 | 18 (25.0) | 19 (20.9) | 21 (24.1) | 58 (23.2) |
| 5–6 | 11 (15.3) | 9 (9.9) | 10 (11.5) | 30 (12.0) |
| More than 6 | 4 (5.6) | 13 (14.3) | 6 (6.9) | 23 (9.2) |
| **English literacy**[*] | | | | |
| Yes | 39 (54.2) | 46 (50.5) | 57 (65.5) | 142 (56.8) |
| No | 33 (45.8) | 45 (49.5) | 30 (34.5) | 108 (43.2) |
| **Education** | | | | |
| None/incomplete primary | 17 (23.6) | 23 (25.3) | 11 (12.6) | 51 (20.4) |
| Primary/Secondary | 30 (41.7) | 35 (38.5) | 46 (52.9) | 111 (44.4) |
| Higher | 3 (4.2) | 3 (3.3) | 5 (5.7) | 11 (4.4) |
| Islamic or other | 22 (30.6) | 30 (33.0) | 25 (28.7) | 77 (30.8) |
| **Address** | | | | |
| Combo/Banjul/Kanifing | 37 (51.4) | 35 (38.5) | 82 (94.3) | 154 (61.6) |
| West Coast | 20 (27.8) | 55 (60.4) | 5 (5.7) | 80 (32.0) |
| Provinces/'Up-river' | 15 (20.8) | 1 (1.1) | 0 (0.0) | 16 (6.4) |
| **Structure of house** | | | | |
| Brick and tiles | 9 (12.5) | 8 (8.8) | 3 (3.4) | 20 (8.0) |
| Mud/sand and corrugate | 14 (19.4) | 14 (15.4) | 3 (3.4) | 31 (12.4) |
| Cement and corrugate | 49 (68.1) | 69 (75.8) | 81 (93.1) | 199 (79.6) |
| **Tribe** | | | | |
| Mandinka | 24 (33.3) | 37 (40.7) | 33 (37.9) | 94 (27.6) |
| Fula | 18 (25.0) | 17 (18.7) | 20 (23.0) | 55 (22.0) |
| Wolof | 13 (18.1) | 15 (16.5) | 20 (23.0) | 48 (19.2) |
| Other | 17 (23.6) | 22 (24.2) | 14 (16.1) | 53 (21.2) |
| **Admission Characteristic** | | | | |
| **Referred**[*] | | | | |
| No | 22 (30.6) | 54 (59.3) | 84 (96.6) | 160 (64.0) |
| Yes | 50 (69.4) | 37 (40.7) | 3 (3.4) | 90 (36.0) |

*(Continued)*

**Table 1.** (Continued)

| Background Characteristic | Hospital 1 (n = 72) | Hospital 2 (n = 91) | Hospital 3 (n = 87) | Total (n = 250) |
|---|---|---|---|---|
| | No. (%) | No. (%) | No. (%) | No. (%) |
| **High risk[c]** | | | | |
| Yes | 47 (65.3) | 45 (50.6) | 11 (12.6) | 103 (41.5) |
| No | 25 (34.7) | 44 (49.4) | 76 (87.4) | 145 (58.5) |
| *Missing* | | 2[d] | | 2[d] |
| **Number of previous contacts*** | | | | |
| 1 to 3 | 21 (29.6) | 30 (33.3) | 27 (31.0) | 78 (31.5) |
| More than 3 | 50 (70.4) | 60 (66.7) | 60 (69.0) | 170 (68.5) |
| *Missing* | 1[d] | 1[d] | | 2[d] |

* Denotes that the variable was entered into the regression analysis.

a. Other occupations include: farmer, student, tailor, civil servant (e.g. police).

b. Number of children in mothers' family, not including current pregnancy/baby born on that admission.

c. High-risk is defined as; multi-pregnancy, pre-eclampsia or pregnancy induced hypertension, severe anaemia, previous C-section/forceps/ventoux delivery and past medical history of diabetes or heart condition (and age <14 years but not applicable). High parity was also considered a risk factor by some of the staff but this was not consistent or featured in guidelines and so was not included (S3 Text).

d. Number of missing data points where information was not available. Valid percentages have been calculated from available information.

**Maternity cards.**   The standard government issue yellow women-held maternity card (Figs 1 and 2) was brought by 94.0% (235/250, 90.3–96.3%) of women and a further 1.2% (3/250, 0.4–3.5%) brought an alternative maternity card e.g. from a private clinic. Of the 238 cards, the content of 2 could not be assessed. However, 80.1% (189/236) were incomplete with at least one unfilled feature; 26.7% (63/236) noted the 'Estimated Date of Delivery' [Table 2]. The maternity card was brought by 94.2% (97/103) of high-risk women, but only 29.9% (29/97) had their status recorded as high-risk on their card (rose to 36.4% [36/97] when 'other document' contents were included) [Table 2]. When risk-status was available on the card, it was normally part of the obstetric history section or written on the top of the card in red (Figs 1 and 2) as there was no designated space for this information. Although accuracy could not be assessed for all fields, evident inaccuracies were also noted; for example, 3% (7/236) of maternity cards had the wrong age recorded based on mothers' reports to our data collectors.

**Referral sheets.**   Referrals (36%, 79/250) were generally emergency cases when a standardised referral sheet, issued by the Ministry of Health, was expected to be brought in addition to the maternity card: 59.5% (47/79) of referred women had a structured referral sheet (Table 2). 81.1% (9/11) of those referred from another hospital carried a referral sheet, compared with only 58.6% (34/58) from a health centre. Of the referral sheets brought, the content of 3 could not be assessed. None of the referral sheets carried all the minimum criteria for safe maternal handover, although 93.2% (41/44) included the 'reason for referral'. Despite a referral sheet providing a place to mark if the case was emergency/non-emergency, only 38.6% (17/44) had this information completed.

**Ultrasound scanning and estimated date of delivery.**   Most (82%, 205/250) women had attended at least one scan to provide accurate estimate of the date of delivery. Results were routinely recorded on an ultrasound report sheet, not the maternity card. However, only 58.8% (100/170) of women who reported attending scanning brought an ultrasound scan report to hospital (35 women who had attended scanning were not able to be assessed for ultrasound scan presence in their documentation due to a researcher error in data collection).

**Fig 4. Structured referral sheet.** Each hospital (and many of the health centres that referred to the hospitals) had centre-specific sheets designed for external referral. All had similar section headings, but documents were not standardised by the government (unlike the maternity card).

**Minimum criteria fulfilment.** When all individual documents were combined, 24.4% (61/250) of women brought documentation that overall met all nine minimum WHO referral criteria. The median score was 8 out of 9 (IQR 7–9) and 68% (170/250) of women had at least 8 of the criteria fulfilled. Estimated delivery date was the least well-fulfilled criteria (64.8% of respondents lacked it; 162/250). Of all documents, maternity cards had the highest median criteria score and were the only document that provided all 9 minimum criteria, however only 15.7% (37/236) of the cards achieved this (S1 Fig).

Minimum criteria scores were categorised into scores below 8 (insufficient, 32.1% of respondents, 80/250) or 8 and above (sufficient 67.9%, 170/250). 'Sufficient' was not defined as 9/9 criteria in the analysis because of limited numbers meeting all 9 criteria (61/250). In logistic regression analysis, being literate in English (OR 2.04 [95% C.I. 1.08–3.85]), having 1–4 children compared to having fewer or more (OR 4.4 [95% C.I. 1.04–18.07]), having more than 3 contacts with healthcare during pregnancy (OR 2.16 [95% C.I. 1.15–4.03]) were all positively

**Table 2. Contents of women-held documents at admission.**

| Document Content Item | Any Document N = 250[b] No. (% of cases) | Maternal Card N = 236[c] No. (% of cards) | Referral Sheet N = 44[d] No. (% of sheets) | Other documents [a] N = 99 No. (% of 'other's) |
|---|---|---|---|---|
| *MINIMUM CRITERIA* | | | | |
| *1. Mother's name* | **242 (96.8)** | 236 (100) | 44 (100) | 97 (98.0) |
| *2. Age* | **241 (96.4)** | 235[e] (99.6) | 34 (77.3) | 74 (74.7) |
| *3. Address* | **238 (95.2)** | 231 (97.9) | 39 (88.6) | 51 (51.5) |
| *4. Parity* | **231 (92.4)** | 230 (97.5) | 4 (9.1) | 19 (19.2) |
| *5. Estimated day of delivery* | **88 (35.2)** | 63 (26.7) | 1 (2.3) | 34 (34.3) |
| *6. Complications in antenatal period [f]* | **241 (96.4)** | 209 (88.6) | 9 (20.5) | 31 (31.3) |
| *7. Relevant past obstetric complications [g]* | **214 (85.6)** | 209 (88.6) | 17 (38.6) | 3 (3.0) |
| *8. Treatments/tests applied thus far* | **217 (86.8)** | 202 (85.6) | 24 (54.5) | 96 (97.0) |
| *9. Results of treatment/tests* | **194 (77.6)** | 165 (69.9) | 21 (47.7) | 92 (92.9) |
| Problem referred for | **60 (24.0)** | 20 (8.5) | 41 (93.2) | 5 (5.1) |
| Recommended place of delivery | **97 (38.8)** | 97 (41.1) | 2 (4.5) | 65 (65.7) |
| Gravida | **214 (85.6)** | 213 (90.3) | 6 (13.6) | 15 (15.2) |
| HIV status | **4 (1.6)** | 2 (0.8) | 0 (0.0) | 2 (2.0) |
| Emergency/risk status | **52 (20.8)** | 38 (16.1) | 17 (38.6) | 2 (2.0) |
| Medications | **242 (96.8)** | 212 (89.0) | 23 (52.3) | 14 (14.1) |
| Contraception | **239 (95.6)** | 200 (84.7) | 2 (4.5) | 0 (0.0) |
| Detail is illegible | **37 (14.8)** | 29 (12.3) | 6 (13.6) | 11 (11.1) |
| Appears incomplete (at least 1 doc) | **199 (79.6)** | 189 (80.1) | 20 (45.5) | 28 (28.3) |
| - 2 documents | **33 (13.2)** | | | |
| - 3 documents | **2 (0.8)** | | | |

a. Combination of all other documents (including ultrasound reports, lab requests, child health booklets, discharge cards, prescription notes and miscellaneous)

b. Percentages were calculated using non-missing data as the denominator (N)

c. 2 antenatal cards were brought by women but unable to be assessed for content and completeness

d. 4 referral sheets were brought by women but were unable to be assessed for content and completeness

e. 7 maternity cards had the wrong age recorded (3%)

f. 'Complications in antenatal period' was regarded as completed if there was any information regarding antenatal history. On the maternity card, this would mean it should always have at least a single entry of the antenatal check-up where the woman had been issued with the card.

g. 'Past obstetric complications' was regarded as any information regarding obstetric history. On the maternity card, this could be left empty if the woman was primiparous. However, it was regular practice for staff to have written 'n/a' or 'primi' to indicate this, which we took to be desirable practice.

and significantly associated with minimum criteria fulfilment. Travelling further than 1 hour to get to hospital (OR 0.34 [95% C.I. 0.15–0.74]) and attending a hospital other than the tertiary referral centre (Hospital 2 OR 0.45 [95% C.I. 0.19–1.02], Hospital 3 OR 0.22 [95% C.I. 0.08–0.60]) were negatively associated with minimum criteria fulfilment (Table 3).

## Qualitative results

SSIs were conducted with 21 members of the multidisciplinary teams in all hospitals (8 doctors, 8 midwives and 5 nurses). Two FGDs were conducted, one consisting of five midwives and the other consisting of four nurses, in one of the three hospitals. The demographic details of the participants interviewed are shown in S2 Table.

Three themes describing facilitators and barriers to effective handover at admission using women-held documents were identified: women as agents for transporting information and

**Table 3. Results of logistic regression analyses exploring associations between women's characteristics and the likelihood of their documentation containing at least 8 out of 9 minimum criteria.**

| Independent Variable/Characteristic | Unadjusted | | Adjusted | |
|---|---|---|---|---|
| | OR (95% CI) | P-value | OR (95% CI) | P-value |
| **Age of mother** | | | | |
| Under 20 | 1[a] | 0.31 | 1[a] | 0.052 |
| 21–29 | 0.61 (0.27–1.37) | | 0.28 (0.10–0.78) | |
| 30 and over | 0.53 (0.23–1.20) | | 0.36 (0.12–1.09) | |
| **Occupation of mother** | | | | |
| Housewife | 1[a] | 0.516 | 1[a] | 0.098 |
| Retail | 0.79 (0.40–1.56) | | 0.68 (0.31–1.47) | |
| Other | 0.67 (0.33–1.34) | | 0.40 (0.17–0.94) | |
| **Time taken to get to hospital** | | | | |
| Under 1 hour | 1[a] | 0.039 | 1[a] | 0.007 |
| 1 hour and above | 0.53 (0.29–0.97) | | 0.34 (0.15–0.74) | |
| **Number of children** | | | | |
| 0 | 1[a] | 0.046 | 1[a] | 0.13 |
| 1–5 | 3.58 (1.09–11.79) | | 4.40 (1.04–18.07) | |
| 5 or more | 2.13 (0.60–7.62) | | 3.81 (0.76–19.06) | |
| **English literacy[b]** | | | | |
| Illiterate | 1[a] | 0.22 | 1[a] | 0.029 |
| Literate | 1.88 (1.10–3.21) | | 2.04 (1.08–3.85) | |
| **Referred for care[b]** | | | | |
| No | 1[a] | 0.429 | 1[a] | 0.741 |
| Yes | 1.25 (0.71–2.20) | | 0.88 (0.40–1.93) | |
| **Number of contacts with healthcare throughout pregnancy** | | | | |
| 3 or less | 1[a] | 0.004 | 1[a] | 0.016 |
| More than 3 | 2.26 (1.29–3.96) | | 2.16 (1.15–4.03) | |
| **Hospital** | | | | |
| 1 | 1[a] | 0.11 | 1[a] | 0.013 |
| 2 | 0.53 (0.26–1.06) | 0.074 | 0.45 (0.19–1.02) | |
| 3 | 0.49 (0.24–1.00) | 0.048 | 0.22 (0.08–0.60) | |

a. Categories of predictor variables that received ORs of 1.00 are reference categories

b. Correlation matrices revealed strong multi-collinearity between 'referred for care', 'brought by ambulance', and 'high-risk'. Therefore referral was selected as the most appropriate variable for the model as it is the most likely of the three to have an impact on the document type and completeness. 'English literacy' and 'education' were also associated and literacy was selected for the same reason.

documents; role of individual healthcare professionals' actions; system and organisational culture. These themes are presented in Table 4 along with sub-themes and supporting quotations.

The code letter after each quotation refers to the cadre of HCP; doctor (D), midwife (M) and nurse (N).

The first theme groups the barriers and facilitators of women as agents for transporting information and documents. Participants reported that whilst women do normally bring documents (itself a facilitator), some individuals arrive with no documents. One doctor explained that this is often due to the woman never having attended an antenatal appointment before.

**Table 4. Qualitative results support quotations of barriers and facilitators to effective handover on admission through use of documentation.**

| Theme | Barrier/ facilitator | Sub-theme | Quotations |
|---|---|---|---|
| • **Women as agents for transporting information and documents** | **Facilitator** | • Women normally bring documents | "*most of our patients, er when they come back for deliveries, their delivery or any admission or come in for any visits, they come back with their antenatal card, previous investigations like the blood investigations, urine and ultrasound scanning*" (D5) |
| | **Barrier** | • Losing smaller sheets e.g. prescription notes so information not in the card is not available | "*Sometimes the patient will miss their prescription because it's a small sheet, they misplace it and they cannot report to you what medications they take*" (D5) |
| **Role of individual HCPs for provision of information and documents** | **Facilitator** | • If everything is written clearly on the maternity card, it can aid handover | "*If everything is written on the* [antenatal] *card, the treatment and the date of the visit and not only on the prescription part, then it will help a lot.*" (D5) |
| | | • Women not able to understand the medical terms (low health literacy) making written notes more important | "*Sometimes the patients, we will ask them but not all of them are able to speak. Not all of them are able to say or understand the medical terms, the medical issues but from the paper we can cross-check and say oh these things have happened*" (N2) |
| | | • Improvisation of HCPs for highlighting high-risk patients in documentation | "*from the clinic, they will just put high-risk on the* [maternity] *card and why. Some will be high parity, some will be pre-eclampsia, high BP.*" (M3) |
| | **Barrier** | • Not enough information written on referral sheet | • "*you don't have enough information in that referring sheet. . . Most of the time. . . the referring doctors or referring nurses that are in other health facilities don't write enough information.*" (D1)<br>• "*if it's a referral, they now bring their antenatal card. . . sometimes the documentations are not enough* [information], *most times they're not enough.*" (D7)<br>• "*the referral notes, mostly they don't have enough information. Sometimes, they don't have the contact of the referring officer. . . Sometimes what-all that is done for the patient mostly are not there, sometimes even the vitals sometimes they miss it.*" (N5) |
| | | • Lack of clarity | "*some people will just put high-risk-hospital delivery* [on the maternity card] *but they will not say why.*" (M3) |
| | | • Illegible handwriting | • "*Most of the time, you need to clerk again because the referring doctors or referring nurses that are in other health facilities don't write enough information, or some don't even write legibly for you to be reading*" (D1)<br>• "*their handwriting is bad . . . Handwriting is important because you're writing for someone to read so if you're writing it and someone else can't read it so it's useless, it's like don't write.*" (D7) |
| | | • Inaccurate written information for referral | "*for example weekends er Fridays are mostly the days that we have er-most of the days that we sometimes have a lot of referrals because they want to empty their hospitals so they can enjoy their weekend. . . sometimes even it's-they write things that are not even happening*" (D7) |
| | | • Lost information on small pieces of paper | "*every investigation is attached to the* [maternity] *card. . . sometimes we will miss it because it's a small sheet, they* [patients] *misplace it. That is where the deficit comes.*" (D5) |

(*Continued*)

**Table 4.** (Continued)

| Theme | Barrier/ facilitator | Sub-theme | Quotations |
|---|---|---|---|
| ***System and organisational culture*** | **Facilitator** | • Standardised referral forms | *"the country developed a referral form document which if you are referring a patient for every health centres. . . Even the private clinics, they too. . . have their own referral forms." (M1)* |
| | | • Designated space on referral forms for feedback | *"in the referring form, there is a place where feedback should be given to the previous centre, but like it's not done." (N7)* |
| | | • Communication between referring and receiving health centre facilitated by use of structured cards | *"the referral form. . . it has to be filled accordingly. . . the way you received the patient, what you managed with the patient and why you want to refer it here, you understand, so that you can avoid unnecessary referrals" (D1)* |
| | **Barrier** | • Lack of supervision and reinforcement of structured referral forms | • *"there is a national problem in regards to the communication from the referral centre to the receiving centre. . .you are just here sitting or somewhere busy doing other things and then a patient arrive in a very critical situation. . ." (D3)*<br>• *"you might not even know the number of the referral centre" (M8)*<br>• *"Sometimes, they [the referral notes] don't have the contact of the referring officer." (N5)* |
| | | • Lack of referral guidelines | *"I think still now there is a challenge on that, from the referral centre to the receiving centre. . . we need like a protocol or guideline." (D3)* |
| | | • Poor attitude of staff and organisational culture on filling referral forms and maternity cards | *"everything that you have done for the patient you have to write feedback [on the referral form]. . . some may be lazy to do the documentation part for the feedback." (M6)* |
| | | • Unqualified staff accompanying women being referred to the receiving facility | • *"the ones doing the referrals are not the ones bringing the patient. . . they assign a very junior nurse. . . so they when they come, they just give you the referral form. When you ask, they say I don't know anything." [M12]*<br>• *"the referrals are left with untrained nurses" [M11]* |
| | | • Non-antenatal patients never issued with documents and so poor handover information available | • *"'Where is your antenatal card' 'I've never gone to an ANC' . . . when you see that you know have a big challenge to do." (M8)*<br>• *"sometimes you know they [the women] come and they have no medical papers." (N2)*<br>• *"They [the women] normally have [antenatal] cards. . . but some will stay at home without that, you will just, you will see them on their day of delivery . . . you don't know nothing" (M4)* |

The general view was that when they do bring the documents, the documents are extremely important for care as they facilitated handing over the antenatal information. This was particularly significant since many of the patients cannot explain the medical issues themselves, commonly due to lack of education and health illiteracy. Respondents also described how smaller loose sheets inside the booklet, such as prescription notes, were often misplaced by the women. Missing notes can result in difficulty in deciding on clinical pathways for the healthcare staff upon admission to the hospital, especially if the women themselves are incapable of explaining their health issues.

The second theme considers the role of individual HCPs' actions as barriers and facilitators to effective handover. Participants suggested that when all the information is written on the antenatal card or the referral form rather than on separate pieces of paper as notes inside the card, this 'will help a lot' on arrival for admission. Incomplete documentation resulted in valuable time being wasted as the admitting healthcare staff would often have to conduct extra thorough examinations and guess the emergency cause of referral which can lead to errors, whereas in the presence of a good history or good handover notes, immediate management could begin. Multiple participants highlighted illegible handwriting as a key barrier to documentation use. Lack of detail and clarity about the exact reason for a woman being "high-risk"

was flagged as an issue. Moreover, individual HCPs at health-centre facilities were reported to be referring women inappropriately to ease pressure on their own facilities. This led to inaccurate reasons for referral on the official referral forms in order to justify the referral, which could have detrimental impacts on patient care and safety.

The third theme recognises how system and organisational facilitators and barriers exist with the use of documents on admission. Contributing factors to individual staff inappropriately referring women include both staff shortages and resource scarcity in their own facilities. For high-risk patient transfers, this also led to reliance on unqualified staff accompanying the woman being referred in an ambulance to the receiving facility not being able to explain the problems of the transferred women. This placed greater importance on the content of documents carried by the woman, including her maternity card and her referral form. Some participants suggested that an electronic notes based system connecting all centres could facilitate better use of medical documentation through increased information exchange efficiency between healthcare teams and overcome the issue of illegible handwritten notes.

## Discussion

Both the quantitative and qualitative arms of this study found that the majority of women delivering in the 3 Banjul and Kanifing hospitals were successfully issued with a maternity card during pregnancy that they brought with them to hospital. However, many of the cards and the referral sheets were incomplete and neither regularly met the WHO minimum content criteria for referral; including that one third of women who were high-risk did not have their risk status recorded. Both the card and the referral sheet were often 'inaccurate' and loose sheets tucked inside the card were often lost. The number of contacts with healthcare during pregnancy, distance from the hospital, and a woman's literacy all influence the completeness of her documents.

Our LMIC sample of pregnant women performed similarly–if not better–than high income countries (HICs) on bringing their women-held documents to hospital. For example, in Australia studies and audits have shown 85–93% compared to the 94% of women in the Gambia [33,34]. This supplements previous literature describing women-held document use between antenatal appointments and agrees that women reliably bring their maternity cards[16,22,23].

However, with regards to other documentation carried with the maternity card and completeness of the documents themselves, the situation was not so positive. It was found that the majority of cards were incomplete and HCPs complained of facing difficulties when admitting a woman as a result. Neither the combination of all documents brought by women, nor the maternity card itself, regularly met all the minimum criteria recommended by WHO and other documents only increased the proportion of women bringing minimum criteria by around 10%. To the best of our knowledge this is the first study to explore the completeness of women-held maternity documents in a LIC[18,19] although the omission of essential clinical information hindering efficient and safe healthcare delivery has been highlighted in previous studies exploring health information exchange in low and middle-income countries[35,36].

To improve patient safety and prevent maternal deaths, it is critical to have available on admission clinical information for high-risk women to prevent delays for essential decision-making and interventions. It is also important for post-natal care to know if the pregnancy had complications or was high-risk. Yet, only 2 in 3 high-risk women had their risk status available in their documentation, and only half of those had it on their maternity card. When risk status was recorded, HCPs stated that it was often unclear why that woman was high-risk. For the high-risk referred women, the incomplete maternity cards became more critical since only half of women referred to hospitals, generally high-risk admissions, carried a referral

sheet with the maternity card. These forms were equally incomplete and HCPs described them as "frequently inaccurate". If these women are also less health literate (unable to "understand the medical terms"), HCPs explained that documentation was more heavily relied upon as they cannot explain their own conditions. As such, we would recommend a new designated section on the maternity cards where high-risk reason could be marked.

Qualitative data confirmed the quantitative evidence of inaccuracies and losses of scan and test results on small pieces of paper usually slipped inside of the card. A further designated space for test results on the maternity card might help overcome this. The WHO's 2018 "Evidence review of home-based maternal records" key-informant data suggested that documents could often be incomplete and inaccurate due to HCPs' views of documentation as an unnecessary task ("double-work") or having illegible handwriting[11]. Similarly, in our study we found participants explaining that HCPs in The Gambia may be 'lazy' with regards to documentation and that illegible handwriting on documents can cause 'time wasting' when trying to clerk the patient on admission.

Qualitative data from HCPs specifically noted that more guidelines and protocols are needed regarding use of referral sheets and filling of maternity cards, which echoes the WHO evidence review suggestion from key informants[11]. If staff could be given training to follow defined guidelines and supervised instructions to write all essential patient data on the same document (e.g. the maternity card), information may be more consistently available. Training could also help motivate behaviour change by explaining the benefits of good documentation and how maternity cards are perceived to be helpful by clinicians.

Significant differences existed between the completeness of documents at each of the hospitals and women who travelled further to get to the hospital had less complete documents. A study of patient held health booklets in Mongolia (for NCDs) had a similar finding[28]. Improved national referral procedures and better national level guidelines for the use of documentation, as suggested by HCPs in this study, could overcome this difference and may help standardise the completeness of documents.

Characteristics of women that were associated with increased completeness of documents included having had more than three antenatal visits; presumably as more frequent reviews offer more opportunity to complete documents. Trials in Thailand, Indonesia and Cambodia have all shown that use of a maternity card is associated with increased antenatal attendance [37–39]. Furthermore, women who were literate were more likely to have complete documentation, as was seen in and investigation of patient-held health booklets in Mongolia[28]. This is potentially a reflection of socioeconomic status, whereby more literate women are likely to attend facilities, demand and receive better care. Therefore, care must be taken to ensure that illiterate women do not continue to receive less complete documents and potentially perpetuate inequalities, since these women are often those with the highest risk of poor outcomes[12].

Limitations of the study included possible recall bias as women were interviewed about their admission at the time of discharge. Misplaced documentation during women's admissions may have underestimated the number of documents, like ultrasound reports, that may have been brought with them during admission but lost during the inpatient period. The 'minimum content' completion of the admission documents may have been over-estimated, as hospital staff may have filled in certain admission sections of the maternal card whilst the woman was on the ward. Since urban hospitals were sampled, the results may be less generalisable to rural Gambia than the urban Banjul area, although all rural areas around Banjul did refer women to these three hospitals. In Pakistan it was shown that maternity cards were more effective in rural than urban centres[40]. On discussion with public health colleagues in the Gambia and reflecting on our study results that show women further from the hospitals had less complete documents, we would expect document use and completeness might be lower. This may

be due to differing resource levels at the rural hospitals and clinics that might place more time-pressure on clinicians.

The strength of this study was that by employing mixed-methods, it enabled us to provide a more complete and comprehensive commentary. Not only have we been able to comment on the completeness of documents, but we have also been able to go some way to explain reasons for why documents were not complete. We included all maternity hospitals in the capital city of Banjul and thus did not have any sampling biases at the level of institutions.

Future studies could include rural areas and referring primary health centre facilities to increase generalisability and to fully understand the perspective of both ends of the referral system. Longitudinal studies that consider maternal outcomes relationship with document completeness would be challenging but important to investigate the importance of continuity of care to patient outcomes and to promote quality improvement interventions. Further research could also investigate opinions of the women themselves about the use of women-held documentation by HCPs in maternity care in LMIC.

## Conclusion

We found that in The Gambia's capital city Banjul and surrounding Kanifing region maternity hospitals, mothers universally carried their maternity cards and all health care providers referred to them. While the recommendations behind women-held documents in maternity services are clear[11–16], the finding that simple low-cost steps to improve the information recorded on the documents could be important to consider for this and other resource limited settings. Simple adaptations to the maternity card (such as spaces for test results and risk-status) and their better completion could capitalise on their almost universal use by women and staff to improve continuity of care and safer births. This is particularly important for high-risk deliveries if referral forms and other documentation continue to be absent or inadequately completed. Similar standardisation of referral forms, alongside national-level guidelines, training, supervision or monitoring for staff may ensure effective completion and maximise use of the cards and referral forms. Together, this would ensure all essential information is available to provide the smoothest handover to hospital-based care for births and prevent any delays to effective treatment and management of complications.

With WHO standards now emphasising quality of maternity care in hospitals, good handover of clinical information to ensure patient safety is likely to improve maternal outcomes, since handover of information is the cornerstone of patient safety and quality of care services [41]. With further development, women-held documents have the potential to play a greater role in the effective information transfer and referral systems in LICs and could optimise delivery of care and the reduction of global maternal mortality.

## Supporting information

**S1 Fig. Chart showing total and individual document completeness against WHO minimum criteria for referrals.**
(TIF)

**S1 Table. Table of hospital background information.** Antenatal care is provided as part of the Maternal, Child health and Family Planning program (MCHFP) and can take place at a variety of health facilities ranging from mobile health posts and local health centres to the tertiary hospital in Banjul. Some health centres have birth facilities, others only provide antenatal care and tell mothers to go to hospital to deliver. Primary healthcare centres can refer women to any of the three hospitals, normally the closest maternity unit geographically. Women

experiencing complications in hospitals in provinces further inland ('upcountry') are sometimes referred to Hospitals 1 or 2.
(DOCX)

**S2 Table. Table of qualitative participant demographics information.**
(DOCX)

**S3 Table. Table of reasons for admission according to woman's own description.** Answers were not mutually exclusive, respondents could select more than one option—therefore %s do not sum to 100. Answers are as patient described, unprompted.
(DOCX)

**S1 Text. Self-designed questionnaire.**
(DOCX)

**S2 Text. Interview topic guide for FGDs and SSIs.**
(DOCX)

**S3 Text. Definitions of 'High-Risk' and 'Complications'.**
(DOCX)

## Acknowledgments

We would like to acknowledge the directors of all three hospitals included in the study: Dr Ahmadou Lamin Samateh, Chief medical director of Edward Francis Small Teaching hospital, Mr Kebba Manneh, Chief executive officer of Kanifing Hospital and Mr Kebba Mane, Chief executive officer of Bundung Maternal and Child Health hospital; for their kind permission to allow us to conduct the study in their hospitals. We would like to thank Buba Manjang of the Ministry of Health of The Gambia. We would also like to thank our good friends Mariama Badjie and Mamadi Sidibeh who, as interpreters, were invaluable to our data collection. We would also like to acknowledge all members of staff on the POSH BMedSc team and the University of Birmingham, especially Dr Gilles de Wildt for his guidance during the protocol design and Dr Sayeed Haque and Dr Alice Sitch for their statistical expertise and assistance. Thanks again to the Arthur Thomson Trust for their financial assistance and sponsorship of one of the authors. Finally, we would like to thank all the staff in all three hospitals for their cooperation and time.

## Author Contributions

**Conceptualization:** Lotta Gustafsson, Fides Lu, Faith Rickard, Christine MacArthur, Carole Cummins, Ivan Coker, Kebba Mane, Kebba Manneh, Amie Wilson, Semira Manaseki-Holland.

**Data curation:** Lotta Gustafsson, Fides Lu, Faith Rickard, Christine MacArthur, Carole Cummins, Ivan Coker, Kebba Manneh, Semira Manaseki-Holland.

**Formal analysis:** Lotta Gustafsson, Fides Lu, Faith Rickard, Christine MacArthur.

**Investigation:** Lotta Gustafsson, Fides Lu, Faith Rickard, Carole Cummins.

**Methodology:** Lotta Gustafsson, Fides Lu, Faith Rickard, Christine MacArthur, Carole Cummins, Ivan Coker, Amie Wilson, Semira Manaseki-Holland.

**Project administration:** Lotta Gustafsson, Fides Lu, Faith Rickard, Christine MacArthur, Ivan Coker, Kebba Mane, Kebba Manneh, Amie Wilson, Semira Manaseki-Holland.

**Resources:** Lotta Gustafsson, Fides Lu, Christine MacArthur, Ivan Coker, Kebba Mane, Kebba Manneh, Semira Manaseki-Holland.

**Supervision:** Christine MacArthur, Carole Cummins, Ivan Coker, Kebba Mane, Kebba Manneh, Amie Wilson, Semira Manaseki-Holland.

**Validation:** Lotta Gustafsson, Fides Lu, Faith Rickard, Christine MacArthur, Semira Manaseki-Holland.

**Visualization:** Lotta Gustafsson, Fides Lu.

**Writing – original draft:** Lotta Gustafsson, Fides Lu.

**Writing – review & editing:** Lotta Gustafsson, Fides Lu, Faith Rickard, Christine MacArthur, Carole Cummins, Ivan Coker, Kebba Mane, Kebba Manneh, Amie Wilson, Semira Manaseki-Holland.

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
