## [Decision Letter · Decision Letter 0]

4 Aug 2019

PONE-D-19-17140

Is the women-held maternity card enabling continuity of care on admission to maternity services in low-income settings? A mixed methods study in Banjul, The Gambia

PLOS ONE

Dear Senior Clinical Lecturer Manaseki-Holland,

Thank you for submitting your manuscript to PLOS ONE. After careful consideration, we feel that it has merit but does not fully meet PLOS ONE’s publication criteria as it currently stands. Therefore, we invite you to submit a revised version of the manuscript that addresses the points raised during the review process.

We advice you to pay special interest in the comments of reviewer 3 and reviewer 4.

We would appreciate receiving your revised manuscript by Sep 18 2019 11:59PM. To enhance the reproducibility of your results, we recommend that if applicable you deposit your laboratory protocols in protocols.io, where a protocol can be assigned its own identifier (DOI) such that it can be cited independently in the future. For instructions see: http://journals.plos.org/plosone/s/submission-guidelines#loc-laboratory-protocols

We look forward to receiving your revised manuscript.

Kind regards,

Astrid M. Kamperman

Academic Editor

PLOS ONE

Journal Requirements:

1. We note that you have indicated that data from this study are available upon request. PLOS only allows data to be available upon request if there are legal or ethical restrictions on sharing data publicly. For information on unacceptable data access restrictions, please see http://journals.plos.org/plosone/s/data-availability#loc-unacceptable-data-access-restrictions.

Reviewers' comments:

Reviewer's Responses to Questions

**Comments to the Author**

1. Is the manuscript technically sound, and do the data support the conclusions?

Reviewer #1: Yes

Reviewer #2: Partly

Reviewer #3: No

Reviewer #4: Yes

2. Has the statistical analysis been performed appropriately and rigorously? 

Reviewer #1: Yes

Reviewer #2: No

Reviewer #3: No

Reviewer #4: I Don't Know

3. Have the authors made all data underlying the findings in their manuscript fully available?

Reviewer #1: Yes

Reviewer #2: No

Reviewer #3: Yes

Reviewer #4: Yes

4. Is the manuscript presented in an intelligible fashion and written in standard English?

Reviewer #1: Yes

Reviewer #2: Yes

Reviewer #3: No

Reviewer #4: Yes

5. Review Comments to the Author

Reviewer #1: this study is well written and very specific to people in field. one of few research that merge quantitaive and qualitative data.

Reviewer #2: This mixed methods study on improving continuity of maternity care by using maternity card. The specific design was convergent parallel mixed-methods study, of which the quantitative component was the interview questionnaire survey and the qualitative component comprised of 2 focus groups and 21 in-depth interviews of healthcare professionals.

Isn’t the women-held maternity documents already well established? If so, the study should focus on either the feasibility in a local setting or completeness/accuracy of the records but not the efficacy in general. I think the authors already took these points into account as reflected by the design, results, and discussion. Hence, some of the texts (e.g. Line 59-61), and the title, should be revised at the authors’ discretion.

The qualitative results were too brief and did not integrate the quantitative findings as proposed. The strength of the mixed methods design was not utilized as anticipated.

Reviewer #3: Review Comments to the Author:

The topic of maternity cards is an interesting and important one and I applaud the authors for their work. I strongly support mixed methods manuscripts; however, this one tries to cover too much ground, and unfortunately, the quality of the text and analyses were sacrificed as a result. I have concerns about the quality of this study and the manuscript itself, which I describe in detail below.

I suggest that the coauthors regroup and decide on a narrower focus of a potential future manuscript. As a side note, in a future manuscript, I suggest incorporating language around how maternity cards can be empowering to women in taking care of themselves and advocating for their care.

Is the manuscript technically sound, and do the data support the conclusions?

The background section does not adequately cover what is already known about the use of maternity cards. The authors describe this as an efficacy study, but it more closely aligns with an effectiveness study since it takes place in a real-world setting.

The authors do not mention the sampling procedure for women participating in the quantitative proportion of the study. Did researchers interview each woman who was discharged until they reached a certain number of women on that day? On line 162, the authors write that research days were selected “at random and including weekends.” The authors fail to describe the specific procedure for randomization. Ultimately, the methods section needs to be written so that these findings could be reproduced.

Further, on line 204, the authors mention that participant were assigned to FGDs or IDIs solely based on their availability. This raises a concern as these two qualitative methodologies have very distinct and differing purposes. For example, FGDs are often used to understand group norms or group dynamics while an IDI allows for a private, in-depth conversation. Additionally, I would want to know why women were not interviewed to understand their perspectives of the maternity card process. If it was a matter of resources, that is fine, but it would have been good to acknowledge that in the text.

There are too many tables - with much more data than what is covered in the text. There are often not adequate text descriptions of the data in the tables. The discussion and conclusion sections are broad and would need to go deeper with the implications of the research as well as recommendations.

Has the statistical analysis been performed appropriately and rigorously?

Statistical analyses have not been performed appropriately and rigorously. In the presentation of descriptive statistics, there were often differing denominators and it was not always clear why. Confidence intervals were erroneously included when presenting descriptive statistics. For multivariate analyses, adjusted odds ratios were not included. Additionally, the authors do not control for clustering of the hospitals.

Is the manuscript presented in an intelligible fashion and written in standard English?

The language in the manuscript is not always intelligible or well organized. Additionally, the presentation of numbers and percentages are not consistent, which is quite distracting.

Reviewer #4: This is a very well-written paper describing interesting, and alarming, results of a mixed-methods study regarding woman-held documents in maternity care of the Gambia. The abundance of data is logically organized and intelligible, and results highlight specific areas for improvement in the completion of cards and, in turn, potential reductions in maternal mortality for the region. The authors conclusions are sound and suggestions for inclusion of space for risk-status and expected delivery date on maternity cards are fitting of the data and very reasonable recommendations for improving maternal care. However, there are aspects of the methods and discussion that detract from the current impact of this paper. Specifically, the methods used to develop the model require more detail to support their proper interpretation. A clearer description of why variables were included in the regression models is needed. Additionally, the discussion should be expanded to better explain and situate the findings. While this may be the first study to assess completeness of cards, the authors should enrich the discussion with comparisons of their results on rates of women carrying documents and qualitative findings with research in other LMIC.

Specific comments:

Line 96: It is unclear why the authors refer to anecdotal evidence?? Is there not empirically-based data? A reference to anecdotal data does not belong in the abstract.

Line 123: LICs should be spelled out first (i.e. “low-income countries (LICs)”). Line 136 should be changed to just LIC.

Line 127: Missing a “)” after (10).

Figure 1: If possible, please rotate the top image so it is easier to read.

Lines 193-197: The method by which variables were chosen to be included in regression analyses needs to be made clearer. Why were these specific few variables chosen? For example, what does distance from hospital have to do with completion of cards? What were the “appropriate exploratory models”??

Table 2 should have asterisk, like Table 1, indicating which variables were included in the modeling.

Line 247: Shouldn’t this be 79/250 not 251?

Line 334-335: Please expand on the studies referenced here. Did they have similar findings to the present study? Why yes or no?

Lines 354-367: What about the other significant sociodemographic factors? What hypothesis do the authors have for the association between sociodemographics (being a housewife, living close to the hospital, literacy) and card completion?

Lines 361-362: This appears to be the only reference to the qualitative data in the discussion. The authors should expand on how their findings are related to previous research and/or describe more the relation between their quant and qual data.

Line 362: I believe this should be changed to “in our sample, 100%..”. “In fact” seems to indicate that staff always review 100%, which cannot be inferred from this data.

Line 381: Please expand on in what ways you would anticipate the situation to be different in rural Gambia and why.

Line 387: What type of sub-group analyses were under-powered? What analysis of high-risk women was underpowered? Almost half of the sample was high-risk.

Line 392: Mother should be changed to mothers.

6. PLOS authors have the option to publish the peer review history of their article (what does this mean?). If published, this will include your full peer review and any attached files.

Reviewer #1: Yes: Dr Mohamad Al-Tannir

Reviewer #2: Yes: Krit Pongpirul

Reviewer #3: No

Reviewer #4: No

---

## [Author Response · Author response to Decision Letter 0]

9 Nov 2019

1. This study is well written and very specific to people in field. one of few research that merge quantitative and qualitative data.

Thank you, we are glad that the benefits of merging quantitative and qualitative data are seen. 

Maternity care is a high priority for the world health organization and sustainable development goals (as part of SDG 3). Although the sub-topic of maternity cards is specific, they are widely used, especially in low-middle income countries. Furthermore, they have featured in WHO guidelines both in the past and as recent as 2018, despite most of the evidence for them being ‘low-certainty’. Maternity cards are the cornerstone of antenatal care and delivery in Low-Middle income countries due to their facilitation of handover (which is key to patient safety across all specialties and countries). We hope that this study, although specific to the Gambia and maternal health, will be relevant to other low-income countries (LICs) and countries in the region that face similar challenges, especially as there is a limited amount of evidence regarding maternity cards from LICs. The effectiveness and feasibility of the card and some of the problems we identified could be generalised to a much wider number of countries where such research is yet to be done.

2. Isn’t the women-held maternity documents already well established? If so, the study should focus on either the feasibility in a local setting or completeness/accuracy of the records but not the efficacy in general. I think the authors already took these points into account as reflected by the design, results, and discussion. Hence, some of the texts (e.g. Line 59-61), and the title, should be revised at the authors’ discretion.

We agree with this comment that we have not conducted an efficacy study. Therefore have made changes in the wording in our ‘aims’ and throughout the paper to emphasise that the study explores the completeness of documents in a real-world setting. We have also made changes to the title to make this clearer. 

3. The qualitative results were too brief and did not integrate the quantitative findings as proposed. The strength of the mixed methods design was not utilized as anticipated.

We understand that we have missed an opportunity to really make the most of the mixed-methods design. Therefore we have made multiple changes. 

1) The qualitative analysis was reviewed and more supporting quotations and subthemes were added to the qualitative results table 

2) Qualitative results that support the quantitative findings were more clearly signposted in the discussion. Furthermore, some of the qualitative findings were relevant in explaining the results of the rerun of the multivariate analysis of the quantitative data. 

3) Qualitative results were used to help select clinically important variables for the rerun of the logistic regression which has been noted in the methodology section

4) Qualitative literature found in a new literature search and from the WHO 2018 Evidence Review of maternity documents was added to the background. 

4. The topic of maternity cards is an interesting and important one and I applaud the authors for their work. I strongly support mixed methods manuscripts; however, this one tries to cover too much ground, and unfortunately, the quality of the text and analyses were sacrificed as a result. I have concerns about the quality of this study and the manuscript itself, which I describe in detail below.

I suggest that the coauthors regroup and decide on a narrower focus of a potential future manuscript. As a side note, in a future manuscript, I suggest incorporating language around how maternity cards can be empowering to women in taking care of themselves and advocating for their care.

Thank you very much in acknowledging the importance of this topic. As mentioned above in response to comment 3, we realise we have not made the most of the mixed-methods design and so have worked on multiple ways of integrating the findings (see above). 

We recognise that much of the ‘too much ground’ partly stems from lack of integration of the quantitative and qualitative findings. It was also a product of finding that the situation on the ground was far more complex than simply the use of the maternity card, and synthesis of the results regarding ‘other documentation’ and ‘referral sheets’ presented challenges. Through significant reorganisation of tables, renewed analysis and added evidence to the qualitative findings, we have tried to substantially strengthen the foundations of our results. 

Specifically, Table 1 and 2 have been combined and reorganized into the three hospitals rather than referred/not referred. We have converted one of the tables into a visual chart. We have moved the table describing reasons for admission to supplementary materials. We have removed the table discussing ‘missing data’ which confused the interpretation of the results. We have rerun the logistic regression analysis and revised the table accordingly. 

Throughout the revised manuscript we have tried to put focus and emphasis on the fact that the paper is about the completeness of the documents as for the documents to have the desired improved effects on care, they have to be complete. 

Our background section now tries to acknowledge that there is a good existing evidence base for women themselves appreciating the maternity card for the ‘empowerment’ and involvement in their own care. A previous and renewed literature search revealed that the major gap in research was partly on the feasibility of the card and its completeness in a (especially low-income) setting. We have added an explanation that resource and time restraints meant that it was not possible to conduct interviews with women, which might have revealed themes regarding empowerment that were not identified through just HCP interviews. We recognise that the women have an important opinion to voice, but this would be an important future research possibility (and was actually a feature in our concurrent research regarding discharge documentation). The combination of the good existing evidence base, lack of time and resources, and desire to keep the study focused on completeness of the documents, meant that we were not able to incorporate the impact of maternity cards on the women’s sense of empowerment. 

5. The background section does not adequately cover what is already known about the use of maternity cards. The authors describe this as an efficacy study, but it more closely aligns with an effectiveness study since it takes place in a real-world setting.

We agree that this is certainly not an efficacy study. We have changed the title and references to the study type throughout the manuscript accordingly. 

We have expanded our background section to include parts of our literature search, and multiple references, that we were previously required to remove due to work count limits and limits to the number of references allowed for our original submission to PLOS Medicine. Therefore, we hope the background section now adequately covers what is known about the maternity cards. Notably, since the original literature search, a ‘WHO Review of Evidence’ has been published regarding women-held documents in maternity care and this has been commented upon heavily in the new background section and discussion. 

6. The authors do not mention the sampling procedure for women participating in the quantitative proportion of the study. Did researchers interview each woman who was discharged until they reached a certain number of women on that day? On line 162, the authors write that research days were selected “at random and including weekends.” The authors fail to describe the specific procedure for randomization. Ultimately, the methods section needs to be written so that these findings could be reproduced.

We have made appropriate changes to our methodology section to describe more accurately our sampling procedure. 

We attempted to purposively select an even distribution of weekdays and weekends across the three study sites to increase the likelihood of obtaining a representative sample. We then recruited as many women being discharged on that day as possible. We did so by being present on the ward from 9 until 1 as we had information from staff that that was the time period that discharges took place. We then asked the women to wait for us to speak to them before they left. Unfortunately we do not know exactly how many women were missed each day, although we can comment on the total number of discharged and that we approximately sampled a quarter of all discharges from the three hospitals in the period

7. Further, on line 204, the authors mention that participant were assigned to FGDs or IDIs solely based on their availability. This raises a concern as these two qualitative methodologies have very distinct and differing purposes. For example, FGDs are often used to understand group norms or group dynamics while an IDI allows for a private, in-depth conversation.

Initially, we planned semi-structured interviews (SSIs) to obtain in-depth views and experiences which may be difficult to elicit in a group setting and focus group discussions (FGDs) to explore more in-depth group dynamics, agreed behaviours and ways of operation. However, the FGDs proved difficult to arrange due to staff availability; it was seemingly rare for large numbers of staff to be available at the same time. For the SSIs, sampling was much more systematic. Participants were allocated to either FGDs or one-to-one SSIs; no staff participated in both.

^This above paragraph has been included in the qualitative methodology section. 

8. Additionally, I would want to know why women were not interviewed to understand their perspectives of the maternity card process. If it was a matter of resources, that is fine, but it would have been good to acknowledge that in the text.

As has been mentioned above (response 6), because the study objectives were mainly around the feasibility and completeness of the documents, it was considered that it would be more valuable to interview HCPs about the use of the cards and potential reasons why the cards would not be completed accurately. Only HCPs write in the cards and so completeness depends on the HCP not the woman. Additionally, time and resource constraints played a part in the decision. We have added a paragraph to explain this decision in the qualitative methodology section and have added mention of how this would be important for future study in the discussion. 

9. There are too many tables - with much more data than what is covered in the text. There are often not adequate text descriptions of the data in the tables. 

The discussion and conclusion sections are broad and would need to go deeper with the implications of the research as well as recommendations

The tables have been significantly restructured and more description of the tables has been added as footnotes. 

Specifically, Table 1 and 2 have been combined and reorganized into the three hospitals rather than referred/not referred. We have converted one of the tables into a visual chart. We have moved the table describing reasons for admission to supplementary materials. We have removed the table discussing ‘missing data’ which confused the interpretation of the results. We have rerun the logistic regression analysis and revised the table accordingly. 

The discussion section has been expanded to include more of the qualitative findings and to more explicitly explore the findings of the logistic regression analysis. We have tried to emphasise the clear recommendations of the study – to make simple changes to the existing card, standardise the referral documents, include all essential information on the maternity card, and introduce national level guidelines – that we feel are best supported by our evidence. 

10. Statistical analyses have not been performed appropriately and rigorously. In the presentation of descriptive statistics, there were often differing denominators and it was not always clear why. Confidence intervals were erroneously included when presenting descriptive statistics. For multivariate analyses, adjusted odds ratios were not included. Additionally, the authors do not control for clustering of the hospitals.

We have sought further expert statistical advice from two statisticians and efforts have been made to review the presentation of our results. The denominators should now be clearer, and if they have changed due to missing data points, this is now explicitly mentioned in the text or footnotes of the tables. 

Confidence intervals have been removed from the presentation of descriptive statistics, as there inclusion was indeed erroneous. 

The results of both unadjusted and adjusted odds ratios from the logistic regression analysis are presented in table 3. The way in which variables were selected for inclusion in the logistic regression has been carefully described in the re-written methods section. 

As there were only three study sites, ‘hospital’ was included as a fixed predictor variable in the final model to control for its influence. 

11. The language in the manuscript is not always intelligible or well organized. Additionally, the presentation of numbers and percentages are not consistent, which is quite distracting.

Alongside significant restructuring of the tables and the addition of a bar-chart, we hope that the presentation of our data is clearer. We have added descriptions and footnotes to the tables to explain the data in a clearer manner. Significant changes to the text in all sections of our paper have been made and we have tried to make improvements to the clarity of language used. As we have removed the erroneous presentation of confidence intervals in the descriptive data, and have reviewed the denominators, we hope that the numbers are now clearer. We now consistently present a % and n/N for every piece of data stated in the results section. 

12. This is a very well-written paper describing interesting, and alarming, results of a mixed-methods study regarding woman-held documents in maternity care of the Gambia. The abundance of data is logically organized and intelligible, and results highlight specific areas for improvement in the completion of cards and, in turn, potential reductions in maternal mortality for the region. The authors conclusions are sound and suggestions for inclusion of space for risk-status and expected delivery date on maternity cards are fitting of the data and very reasonable recommendations for improving maternal care. However, there are aspects of the methods and discussion that detract from the current impact of this paper. Specifically, the methods used to develop the model require more detail to support their proper interpretation. A clearer description of why variables were included in the regression models is needed.

We have sought further expert statistical advice and the regression analysis has been re-run. Characteristics, in the new analysis, were chosen based on clinical rationale, supported by both literature and the findings from the qualitative arm of the study. If variables were found to be too closely related on examination of cross-tabulations and the correlation matrices resulting in multi-collinearity, then they were excluded and clinical rationale was used to choose the most appropriate variable. This process has been explained in the methods section.

The results of the new regression analysis have been explored far more extensively in the discussion and we linked them to the qualitative findings and the literature to better understand these associations. 

13. Additionally, the discussion should be expanded to better explain and situate the findings. While this may be the first study to assess completeness of cards, the authors should enrich the discussion with comparisons of their results on rates of women carrying documents and qualitative findings with research in other LMIC.

We have added clearer reference to relevant studies into women carrying documents in the second paragraph of the discussion (and we hope throughout). Other literature has also been used to help understand the new regression analysis results. 

14. Line 96: It is unclear why the authors refer to anecdotal evidence?? Is there not empirically-based data? A reference to anecdotal data does not belong in the abstract

We agree with this comment and therefore have removed the sentence from the abstract. A WHO 2018 evidence review that suggests, based on key-informant data, that guidelines, training and support is needed to encourage appropriate use and we have mentioned this in the discussion. 

15. Line 123: LICs should be spelled out first (i.e. “low-income countries (LICs)”). 

Thank you; this has been changed in line with your recommendation. 

16. Line 136 should be changed to just LIC.

Thank you; this has been changed in line with your recommendation.

17. Line 127: Missing a “)” after (10)

Thank you; this has been changed in line with your recommendation.

18. Figure 1: If possible, please rotate the top image so it is easier to read.

Thank you; this has been changed in line with your recommendation.

19. Lines 193-197: The method by which variables were chosen to be included in regression analyses needs to be made clearer. Why were these specific few variables chosen? For example, what does distance from hospital have to do with completion of cards? What were the “appropriate exploratory models”??

The regression analysis has been re-run. Characteristics were chosen based on clinical rationale, supported by both literature and the findings from the qualitative arm of the study. If variables were found to be too closely related following examination of cross-tabulations and the correlation matrices,resulting in multi-collinearity then they were excluded and clinical rationale was used to choose the most appropriate variable to include. This process has been explained in the methods section.

The methods section now contains a clearer description of the statistical methods and “appropriate exploratory models” has been removed. 

20. Table 2 should have asterisk, like Table 1, indicating which variables were included in the modeling.

Thank you; this has been changed in line with your recommendation.

21. Line 247: Shouldn’t this be 79/250 not 251?

Thank you; this has been changed in line with your recommendation.

22. Line 334-335: Please expand on the studies referenced here. Did they have similar findings to the present study? Why yes or no?

The discussion has been significantly restructured to include a more detailed literature review and explicit comparison of the results to the literature. 

23. Lines 354-367: What about the other significant sociodemographic factors? What hypothesis do the authors have for the association between sociodemographics (being a housewife, living close to the hospital, literacy) and card completion?

The results of the new regression analysis (where variables were selected based on clinical rationale from the literature and the qualitative arm of the study) have been explored far more extensively in the discussion and we tried to link them to the qualitative findings and the literature to explain the potential reasons for them. 

24. Lines 361-362: This appears to be the only reference to the qualitative data in the discussion. The authors should expand on how their findings are related to previous research and/or describe more the relation between their quant and qual data

We have re-written a lot of the discussion section with heavy emphasis now placed on reconciling the quantitative and qualitative findings to build a picture. We have especially made use of the rerun of the logistic regression analysis to relate our findings to previous literature. 

We have also rerun our literature review, which helpfully highlighted studies we had not previously commented upon. 

25. Line 362: I believe this should be changed to “in our sample, 100%..”. “In fact” seems to indicate that staff always review 100%, which cannot be inferred from this data.

Thank you; this point has been removed from the discussion as it was considered to not be clear. Greater discussion of the qualitative findings has replaced it. 

26. Line 381: Please expand on in what ways you would anticipate the situation to be different in rural Gambia and why.

Following previous discussions we have had with colleagues in The Gambia alongside added evidence provided by key-informants in the most recent WHO review of evidence on maternity cards, we have tried to expand on how the situation may be different in rural Gambia in the discussion of the limitations of the study. We make particular parallels with a study in Pakistan.

Since urban hospitals were sampled, the results may be less generalisable to rural Gambia than the urban Banjul area, although all rural areas around Banjul did refer women to these three hospitals. In Pakistan it was shown that maternity cards were more effective in rural than urban centres(40). On discussion with public health colleagues in the Gambia and reflecting on our study results that show women further from the hospitals had less complete documents, we would expect document use and completeness might be lower. This may be due to differing resource levels at the rural hospitals and clinics that might place more time-pressure on clinicians. 

27. Line 387: What type of sub-group analyses were under-powered? What analysis of high-risk women was underpowered? Almost half of the sample was high-risk.

Thank you; the sentence in question was in reference to analysis performed in previous drafts that was not included in the final publication. Therefore it has been removed. 

28. Line 392: Mother should be changed to mothers. 

Thank you; this has been changed in line with your recommendation. 

We have altered the formatting of the document in line with the information in the links you have provided. We have separated the supplementary materials and figures from the file, and labeled them appropriately.

---

## [Decision Letter · Decision Letter 1]

18 Dec 2019

PONE-D-19-17140R1

The content and completeness of women-held maternity documents before admission for labour; a mixed methods study in Banjul, The Gambia

PLOS ONE

Dear Senior Clinical Lecturer Manaseki-Holland,

Thank you for submitting your manuscript to PLOS ONE. After careful consideration, we feel that it has merit but does not fully meet PLOS ONE’s publication criteria as it currently stands. Therefore, we invite you to submit a revised version of the manuscript that addresses the points raised during the review process.

Both reviewers raised some remaining textual issues, and some minor points regarding the interpretation of your findings.  

We would appreciate receiving your revised manuscript by Feb 01 2020 11:59PM. To enhance the reproducibility of your results, we recommend that if applicable you deposit your laboratory protocols in protocols.io, where a protocol can be assigned its own identifier (DOI) such that it can be cited independently in the future. For instructions see: http://journals.plos.org/plosone/s/submission-guidelines#loc-laboratory-protocols

We look forward to receiving your revised manuscript.

Kind regards,

Astrid M. Kamperman

Academic Editor

PLOS ONE

Reviewers' comments:

Reviewer's Responses to Questions

**Comments to the Author**

1. If the authors have adequately addressed your comments raised in a previous round of review and you feel that this manuscript is now acceptable for publication, you may indicate that here to bypass the “Comments to the Author” section, enter your conflict of interest statement in the “Confidential to Editor” section, and submit your "Accept" recommendation.

Reviewer #3: All comments have been addressed

Reviewer #4: (No Response)

2. Is the manuscript technically sound, and do the data support the conclusions?

Reviewer #3: Yes

Reviewer #4: Yes

3. Has the statistical analysis been performed appropriately and rigorously? 

Reviewer #3: Yes

Reviewer #4: Yes

4. Have the authors made all data underlying the findings in their manuscript fully available?

Reviewer #3: Yes

Reviewer #4: No

5. Is the manuscript presented in an intelligible fashion and written in standard English?

Reviewer #3: Yes

Reviewer #4: Yes

6. Review Comments to the Author

Reviewer #3: This revised manuscript is a marked improvement from the original submission. It is much clearer and is synthesized very well. I only have a few outstanding concerns to be addressed:

1) I couldn't find S1 appendix and perhaps this information is already included there, but could the authors include the number of participants and the cadre of participants in the FGDs as they did for the SSIs in the body of the text?

2) I have a few comments on table 1. The number of children includes 6 twice- "5-6" and "6 or more." Also, the number of previous contacts includes 3 twice- "1 to 3" and "3 or more." Additionally, I don't fully understand the difference between the "1d" and "2d" notations. Can the authors clarify the difference between these? Also, what does "2d2d" in the high-risk section mean?

3) In the results section, lines 343-347, it would be useful to add odds ratios and specify what variables were significant in unadjusted and adjusted models.

4) In table 3, I'm not sure why/how the reference categories have p values. For example, for a categorical variable with 3 categories or levels, the p values would be looking at statistical significance comparing the 1st and 2nd categories/levels and the 1st and 3rd categories/levels, so there should only be p values for the 2nd and 3rd categories.

Reviewer #4: I commend the authors for a thorough response to my comments and revisions to the paper that greatly improve its readability and impact. The regression and qualitative methods and results are much more clearly explained and presented, and mixed-methods results have been well integrated. Below, I detail some minor remaining issues in the current revision.

Title – I believe you’d want to use a colon not semi-colon.

Line 113-116 – This seems more fitting in the discussion.

Qualitative methods: Traditionally you would want more than one coder of transcripts. The authors should explain why they only had one person code all transcripts and what was agreement on the 4 transcripts coded by other researcher?

Table 1:

• Ages seem to not include 20 (under 20 and 21+)

• Something is wrong with the education row. The numbers don’t line up to categories and the last category is Islamic…

• What is 2d 2d in missing high-risk?

Line 269 - It is not clear what documents were considered in this calculation. The authors state they were “not including loose scan or test result sheets” but then seem to consider these as “less frequently presented documents (ultrasound reports, prescription notes, scraps of 272 paper, child health reports, miscellaneous lab requests/results)”. What is a loose scan vs an ultrasounds, for example?

Line 282 – Why couldn’t the content of the card be assessed?

Line 301 – Why couldn’t the content of the referral be assessed?

Figure 4 does not make sense to me. I don’t understand the y axis, did women have hundreds of documents? This does not seem the best way to illustrate the point made in the text, that only the maternity card had met all 9 criteria.

Line 344 – Why don’t the authors mention the significant association of numbers of contacts throughout pregnancy in the results and discussion intro paragraph?

Table 3. I am unsure of what the p-value next to the reference category would mean. What is this a comparison of?

Table 4. [B] and [F] unnecessary considering the bolded row headings of barrier and facilitator.

Line 452 - It seems that the qualitative data DOES support this (quote from M6 in org barriers). The authors might want to reflect on how this was an issue identified here, so perhaps training should emphasize how helpful clinicians find the cards as a motivator. Or, perhaps a different incentive to complete them is needed.

Line 510 – ALMOST universally.

7. PLOS authors have the option to publish the peer review history of their article (what does this mean?). If published, this will include your full peer review and any attached files.

Reviewer #3: No

Reviewer #4: No

---

## [Author Response · Author response to Decision Letter 1]

1 Feb 2020

1. I couldn't find S1 appendix and perhaps this information is already included there, but could the authors include the number of participants and the cadre of participants in the FGDs as they did for the SSIs in the body of the text?

The S1 appendix provides some information about each of the hospital study sites, including the total number of nurses, midwives and doctors working at each site. We will make sure that this appendix is uploaded successfully on our next submission. 

To address your point we have tried to clarify the wording of this sentence/paragraph to make it clearer what cadre of participants were in each of the FDGs. 

2. I have a few comments on table 1. The number of children includes 6 twice- "5-6" and "6 or more." Also, the number of previous contacts includes 3 twice- "1 to 3" and "3 or more." Additionally, I don't fully understand the difference between the "1d" and "2d" notations. Can the authors clarify the difference between these? Also, what does "2d2d" in the high-risk section mean?

Apologies, these were errors when reconfiguring the table from the SPSS data – it was meant to say “more than 6” and “more than 3”, thank you for spotting it. 

Apologies, the “2d2d” was another formatting mistake and has been corrected. Hopefully this now makes the notation easier to understand. 

The 1d and 2d shows where missing values were in each subgroup – i.e. in high-risk, two women from hospital 2 had a their risk status missing from their questionnaire sheet, therefore in the total column two women had this data point missing. In ‘number of previous contacts’, one participant from hospital 1 and one participant from hospital 2 had this piece of data missing from their questionnaire sheets, which totals two women missing this data (the far right column). 

Due to these mistakes we went back over the table to ensure all data was correct and have made a few appropriate changes. We have also changed Supplementary Material 1 due to the mislabelling of the hospital numbers. 

3. In the results section, lines 343-347, it would be useful to add odds ratios and specify what variables were significant in unadjusted and adjusted models.

Thank you – we have added these to the paragraph (please see below). 

In logistic regression analysis, being literate in English (OR 2.04 [95% C.I. 1.08-3.85]), having 1-4 children compared to having fewer or more (OR 4.4 [95% C.I. 1.04-18.07]), having more than 3 contacts with healthcare during pregnancy (OR 2.16 [95% C.I. 1.15-4.03]) were all positively and significantly associated with minimum criteria fulfilment. Travelling further than 1 hour to get to hospital (OR 0.34 [95% C.I. 0.15-0.74]) and attending a hospital other than the tertiary referral centre (Hospital 2 OR 0.45 [95% C.I. 0.19-1.02], Hospital 2 OR 0.22 [95%C.I. 0.08-0.60]) were negatively associated with minimum criteria fulfilment [Table 3].

4. In table 3, I'm not sure why/how the reference categories have p values. For example, for a categorical variable with 3 categories or levels, the p values would be looking at statistical significance comparing the 1st and 2nd categories/levels and the 1st and 3rd categories/levels, so there should only be p values for the 2nd and 3rd categories.

The p value against the reference category is the p value for the variable’s overall contribution to the model (as are the p values for binary variables). As the p values against other categories do not add information above that provided by the 95% Cis, they are redundant and have been removed from the table (but could be reinstated if the editor wishes). 

5. Title – I believe you’d want to use a colon not semi-colon.

Thank you, we have now changed this in line with your recommendation. 

6. Qualitative methods: Traditionally you would want more than one coder of transcripts. The authors should ex

plain why they only had one person code all transcripts and what was agreement on the 4 transcripts coded by other researcher?

It is considered good practice in qualitative research for more than one research to be involved in coding transcripts but there is no absolute requirement for all transcripts to be coded by another researcher (see for example N Gale et al BMC Res Methods 2013 where it is suggested that another researcher code four transcripts). The concept of interrater agreement is contested in qualitative research (C Pope BMJ 2000), and in this study the other coder contributed to the development of coding, improving the quality of the analysis. Time and resource precluded the double coding of all transcripts and a comment to that effect could be added if the editor wishes.

7. Table 1:

• Ages seem to not include 20 (under 20 and 21+)

• Something is wrong with the education row. The numbers don’t line up to categories and the last category is Islamic…

• What is 2d 2d in missing high-risk?

- We have changed this to ’20 and under’, thank you for spotting this mistake. 

- The lining up of Table 1 when opened on a windows computer was flawed. We have now edited the table on a windows computer to make sure that the numbers line up. ‘Islamic’ category refers to home or small group based education about the Quran, and the student does not learn to read or write English which is the written language or commerce, business and general use. However as the person has exposure to some education we felt it is important to keep this category separate. We have inserted a foot note to say: renamed this as informal/formal Quranic education without learning to read English the national language

- This was a typo and has been correcting, thank you for spotting it. 

8. Line 269 - It is not clear what documents were considered in this calculation. The authors state they were “not including loose scan or test result sheets” but then seem to consider these as “less frequently presented documents (ultrasound reports, prescription notes, scraps of 272 paper, child health reports, miscellaneous lab requests/results)”. What is a loose scan vs an ultrasounds, for example?

Thank you very much for making us aware of this statement that we agree lacks clarity. We recognised that the calculation also features in our abstract, so we returned to our raw data to ensure that our statement was accurate. The phrase ‘loose sheets’ remained there after an earlier draft and needed review. 

We decided that to ensure clarity, we have calculated the value as ‘antenatal cards or referral sheets’ and now explain this clearly both in the paragraph and the abstract. We have made this decision to not include ‘less frequently presented documents’ (which were often ‘loose’) for two reasons. Firstly, these were the only two documents to include significant clinical information (the less frequently presented documents were therefore considered less clinically important to comment on when considering if a woman had presented with any documentation). Secondly, there was a data collection error where 40 participants did not have the presence ultrasound/test requests recorded. This was alluded to on line 323 (“35 women who had attended scanning were not able to be assessed for ultrasound scan presence in their documentation” – 5/40 did not attend scanning). We have added explanation of this to line 324 to make it clear that it was researcher error. 

With regards to the paragraph you have commented on, one participant presented with no documents, but we are missing data as to whether she brought an ultrasound sheet with her or not. Because of this we cannot comment on the number of women who brought ‘no documentation’ of any kind. If the editors feel further clarity is needed regarding this matter, please let us know. 

9. Line 282 – Why couldn’t the content of the card be assessed?

The content of the cards could not be assessed because although the mother could confirm that she had brought an antenatal card with her to hospital, the cards themselves were elsewhere in the hospital e.g. on the Neonatal unit with the baby. 

We have added this to the text. 

10. Line 301 – Why couldn’t the content of the referral be assessed?

The content on some occasions could not be assessed because some of the referral sheets that women recalled bringing with them were elsewhere on the ward rather than with the woman/unable to found by researchers. 

We have added this explanation to the text. 

11. Figure 4 does not make sense to me. I don’t understand the y axis, did women have hundreds of documents? This does not seem the best way to illustrate the point made in the text, that only the maternity card had met all 9 criteria.

We have removed the numbered labels on the Y-axis to make the bars represent the percentage of that type of document that met the minimum criteria (either 9 out 9, or 8 out of 9). We have also altered the X-axis title to simplify the message. 

The minimum criteria fulfilment of ‘referral sheets’ and ‘other documents’ has been added to emphasise that only the maternity card met all 9 criteria. 

We are happy to adapt or remove this figure if the editor wishes. 

12. Line 344 – Why don’t the authors mention the significant association of numbers of contacts throughout pregnancy in the results and discussion intro paragraph?

Thank you, this has now been added (with odds ratios) to the appropriate paragraphs. 

13. Table 3. I am unsure of what the p-value next to the reference category would mean. What is this a comparison of?

Please see response to comment on Table 3 above. 

14. Table 4. [B] and [F] unnecessary considering the bolded row headings of barrier and facilitator.

Thank you, we have changed this in line with your suggestion. 

15. Line 452 - It seems that the qualitative data DOES support this (quote from M6 in org barriers). The authors might want to reflect on how this was an issue identified here, so perhaps training should emphasize how helpful clinicians find the cards as a motivator. Or, perhaps a different incentive to complete them is needed.

This was an important point to have been brought to our attention, thank you. We reflected on this and have added appropriate comments on this in our discussion. 

16. Line 510 – ALMOST universally.

Thank you, we’ve made this change.

---

## [Editor Report · Decision Letter 2]

5 Feb 2020

PONE-D-19-17140R2

The content and completeness of women-held maternity documents before admission for labour: a mixed methods study in Banjul, The Gambia

PLOS ONE

Dear Senior Clinical Lecturer Manaseki-Holland,

Thank you for submitting your manuscript to PLOS ONE. After careful consideration, we feel that it has merit but does not fully meet PLOS ONE’s publication criteria as it currently stands. Therefore, we invite you to submit a revised version of the manuscript that addresses a final point.

Please remove Figure 4 from the text. You might add the figure it to the supplementary material.

We would appreciate receiving your revised manuscript by Mar 21 2020 11:59PM. To enhance the reproducibility of your results, we recommend that if applicable you deposit your laboratory protocols in protocols.io, where a protocol can be assigned its own identifier (DOI) such that it can be cited independently in the future. For instructions see: http://journals.plos.org/plosone/s/submission-guidelines#loc-laboratory-protocols

We look forward to receiving your revised manuscript.

Kind regards,

Astrid M. Kamperman

Academic Editor

PLOS ONE

---

## [Author Response · Author response to Decision Letter 2]

20 Feb 2020

Please remove Figure 4 from the text. You might add the figure it to the supplementary material.

Thank you for your comment regarding Figure 4. We have removed figure 4 and added it to the supplementary materials.

---

## [Editor Report · Decision Letter 3]

21 Feb 2020

The content and completeness of women-held maternity documents before admission for labour: a mixed methods study in Banjul, The Gambia

PONE-D-19-17140R3

Dear Dr. Manaseki-Holland,

We are pleased to inform you that your manuscript has been judged scientifically suitable for publication and will be formally accepted for publication once it complies with all outstanding technical requirements.

With kind regards,

Astrid M. Kamperman

Academic Editor

PLOS ONE
---

## [Editor Report · Acceptance letter]

28 Feb 2020

PONE-D-19-17140R3 

The content and completeness of women-held maternity documents before admission for labour: a mixed methods study in Banjul, The Gambia 

Dear Dr. Manaseki-Holland:

I am pleased to inform you that your manuscript has been deemed suitable for publication in PLOS ONE. Congratulations! Your manuscript is now with our production department. 

With kind regards,

on behalf of

Dr. Astrid M. Kamperman 

Academic Editor

PLOS ONE